

# Prognostic and clinicopathological role of RACK1 for cancer patients: a systematic review and meta-analysis

Qiuhao Wang[1], Sixin Jiang[1], Yuqi Wu[1], You Zhang[1], Mei Huang[1], Yan Qiu[2] and Xiaobo Luo[1]

[1] State Key Laboratory of Oral Diseases, National Clinical Research Center for Oral Diseases, Chinese Academy of Medical Sciences Research Unit of Oral Carcinogenesis and Management, West China Hospital of Stomatology, Sichuan University, Chengdu, China
[2] Laboratory of Pathology, Clinical Research Center for Breast, Department of Pathology, West China Hospital, Sichuan University, Chengdu, China

## ABSTRACT

**Background**. The receptor for activated C kinase 1 (RACK1) expression is associated with clinicopathological characteristics and the prognosis of various cancers; however, the conclusions are controversial. As a result, this study aimed to explore the clinicopathological and prognostic values of RACK1 expression in patients with cancer.

**Methodology**. PubMed, Embase, Web of Science, Cochrane Library, and Scopus were comprehensively explored from their inception to April 20, 2023, for selecting studies on the clinicopathological and prognostic role of RACK1 in patients with cancer that met the criteria for inclusion in this review. Pooled hazard ratios (HRs) and 95% confidence intervals (CIs) were used to assess the prognosis-predictive value of RACK1 expression, while pooled odds ratios (ORs) and 95% CIs were used to evaluate the correlation between RACK1 expression and the clinicopathological characteristics of patients with cancer. The quality of the included studies was evaluated using the Newcastle-Ottawa Scale.

**Results**. Twenty-two studies (13 on prognosis and 20 on clinicopathological characteristics) were included in this systematic review and meta-analysis. The findings indicated that high RACK1 expression was significantly associated with poor overall survival (HR = 1.62; 95% CI, 1.13–2.33; $P = 0.009$; $I^2 = 89\%$) and reversely correlated with disease-free survival/recurrence-free survival (HR = 1.87; 95% CI, 1.22–2.88; $P = 0.004$; $I^2 = 0\%$). Furthermore, increased RACK1 expression was significantly associated with lymphatic invasion/N+ stage (OR = 1.74; 95% CI, 1.04–2.90; $P = 0.04$; $I^2 = 79\%$) of tumors.

**Conclusions**. RACK1 may be a global predictive marker of poor prognosis in patients with cancer and unfavorable clinicopathological characteristics. However, further clinical studies are required to validate these findings.

Corresponding authors
Yan Qiu, qiuyan227@126.com
Xiaobo Luo, xiaobol@scu.edu.cn

## INTRODUCTION

Cancer treatment has advanced significantly; however, cancer remains a serious public health concern, with 608,570 cancer-related deaths reported in 2021 in the United States (*Li et al., 2021*; *Siegel et al., 2021*; *Wang et al., 2022*). One reason for the low five-year survival rate of patients with cancer may be the lack of effective predictors of cancer prognosis (*Emens et al., 2019*). Biomarkers are now widely used in cancer diagnosis, treatment, and prognosis prediction, with these prognostic indicators serving as crucial early intervention indicators, improving the prognosis of patients with cancer (*Anaya et al., 2016*; *Chen et al., 2022*; *Herberts et al., 2022*; *Lee et al., 2021*). As a result, it is necessary to identify biomarkers that can be utilized as prognosticators in patients with cancer.

The receptor for activated C kinase 1 (RACK1) is a highly conserved WD40 repeat protein that acts as a multifunctional scaffold to mediate cellular functions (*Dan et al., 2020*). It was originally identified as a protein anchored by protein kinase C (PKC), with roles in maintaining the stability of active PKC; additionally, it was reported to be ubiquitously expressed in a wide range of normal tissues, such as nervous system and spleen (*McCahill et al., 2002*; *Ron et al., 1994*; *Ron & Mochly-Rosen, 1994*). RACK1 may play versatile roles in various tissues as a scaffold protein; for instance, it can maintain intestinal homeostasis by protecting the integrity and regulating the growth of the intestinal epithelium, as well as mediate the normal development of the nervous system by regulating the Wnt/$\beta$-catenin and Shh pathways in neural stem cells (*Cheng & Cartwright, 2018*; *Cheng, Pai & Cartwright, 2018*; *Yang et al., 2019*). Furthermore, it could trigger cardiovascular disease by modulating the contraction of vascular smooth muscle cells (*Zhu & Jackson, 2017*). Notably, previous studies reported RACK1 to be closely associated with the prognosis of patients with cancer because of its involvement in several tumor-related signaling pathways, such as the Src/FAK (*Ou et al., 2022*), AKT/mTOR (*Zhang et al., 2016*), IKK/NF-$\kappa$B (*Yao et al., 2014*), and Wnt/$\beta$-catenin pathways (*Yu et al., 2021*). In the digestive system, high RACK1 expression is significantly associated with poor prognosis in oral squamous cell carcinoma (OSCC) (*Liu et al., 2018*), esophageal squamous cell carcinoma (ESCC) (*Wang et al., 2015*), and pancreatic cancer (PC) (*Li et al., 2016*); however, several studies have demonstrated that RACK1 acts as a tumor suppressor in gastric cancer (GC) (*Chen et al., 2015*; *Yu et al., 2021*), indicating that it may be an organ-specific tumor marker. For non-digestive cancers, including non-small cell lung cancer (NSCLC), breast cancer (BC), and glioma, RACK1 is a significant biomarker of poor prognosis (*Cao et al., 2010*; *Lv et al., 2016*; *Qu et al., 2017*). Additionally, RACK1 may correlate with poor clinicopathological features, such as lymphatic invasion (*Zhong et al., 2013*), but the opposite was observed in PC (*Zhang et al., 2019*).

Multiple studies have reported that RACK1 is closely related to cancer prognosis; however, its role in cancer prognosis remains controversial. As a result, a systematic meta-analysis is required to better understand RACK1's involvement in cancer prognosis as well as its predictive value, improving clinical decision-making. The purpose of this review was to clarify the prognostic value of RACK1 expression in cancer and its correlation with the clinicopathological characteristics of patients with cancer.

## METHODS

### Protocol and registration

This systematic review with meta-analysis was conducted in compliance with the Preferred Reporting Items for Systematic Reviews and Meta-Analyses (PRISMA) protocol (Table S1). The study protocol was drafted and registered in the International Prospective Register of Systematic Reviews (PROSPERO) database (CRD42022351129).

### Search strategy

Five databases (PubMed, Embase, Web of Science, Cochrane Library, and Scopus) were thoroughly explored from their inception to April 20, 2023, using the following keywords: "receptor of activated c kinase 1" and "neoplasms". Potentially relevant literature was also obtained through manual searches of the reference lists of the included studies. Detailed search strategies are presented in Table S2.

### Inclusion and exclusion criteria

#### Inclusion criteria

The criteria for inclusion of the study were as follows: (1) diagnosis of patients with certain malignant tumors; (2) detection of RACK1 expression in cancer tissues by immunohistochemistry (IHC) and reverse transcription-polymerase chain reaction (RT-PCR); (3) patient stratification according to RACK1 expression; (4) investigation of the relationship between RACK1 and clinicopathological significance or prognosis; and (5) original research on humans.

#### Exclusion criteria

Literature was excluded in the conditions below: (1) the articles were reviews, case reports, letters, abstracts, or comments; (2) insufficient data were available to obtain hazard ratios (HRs) of survival-related and odds ratios (ORs) regarding the correlation of RACK1 expression with clinicopathological features; (3) patients in the according study were less than 50; (4) the cohort being studied is replicated; and (5) literatures were published in languages other than English.

### Study selection and data extraction

Two authors (YQ Wu and SX Jiang) performed a preliminary screening of the literature based on titles and abstracts. Subsequently, two other authors (QH Wang and Y Zhang) evaluated the full-text. Any disagreements were discussed by the review group until a consensus was reached.

Following selection, two authors (QH Wang and Y Zhang) extracted the following data from the included articles: (1) first author; (2) year of publication; (3) country; (4) number of patients; (5) patients' age; (6) detection methods; (7) treatment; (8) follow up; (9) cut-off value; (10) survival information; (11) TNM stage; (12) hazard ratios (HRs) and the 95% confidence intervals (CIs) of RACK1 for survival; (13) clinicopathological characteristics. When HR and 95% CI were not reported, the data were analyzed using survival-related Kaplan–Meier curves according to the method described by *Tierney et al. (2007)*.

## Quality assessment

Two authors (SX Jiang and M Huang) independently used the Newcastle-Ottawa Scale to assess the quality of the cohort studies, and any disagreements were resolved by a third author (*Stang, 2010*). Studies with a score ≥ 7 (score range, 0–9) were considered high quality.

## Statistical analysis

HRs and 95% CIs were used to assess the patient prognosis. HR >1 indicated poor survival in the group with high RACK1 expression, whereas HR <1 indicated poor survival in the group with low RACK1 expression. If the results of both univariate and multivariate Cox regression analyses had been reported, multivariate models were preferred to pool the data. ORs with 95% CIs were used to determine the association between RACK1 expression and clinicopathological characteristics. The heterogeneity of the meta-analysis was tested using Cochran's Q test and the Higgins I-squared statistic ($I^2$). Significant heterogeneity was indicated by $I^2$ >50% or a *P*-value <0.1, and a random effects model was then applied; otherwise, a fixed effects model was preferred. Subgroup, meta-regression, and sensitivity analyses were performed in the presence of significant heterogeneity. Publication bias was evaluated using funnel plots, Begg's test and Egger's test. The level of statistical significance was set at a *P*-value of <0.05. Statistical analysis and visualization were performed using Revman 5.4 and Stata 12.

# RESULTS

## Process of study selection

From the five databases, 2,269 publications were identified, and seven relevant publications were manually searched. After removing 1,332 duplicate studies, 944 publications were eliminated based on their titles and abstracts. The full texts of the remaining 86 papers were reviewed further, and 22 studies were included in the meta-analysis. Of these, 13 studies evaluated the predictive value of RACK1 for cancer prognosis, and 20 studies assessed the correlation between RACK1 expression and clinicopathological features. The search and detailed selection processes are shown in the flowchart (Fig. 1).

## General characteristics of the included studies

The included 22 studies were published between 2010 and 2022, with eleven studies (50%) published in the last five years (*Han et al., 2018*; *Li et al., 2019*; *Liu et al., 2017*; *Liu et al., 2018*; *Qu et al., 2017*; *Shen et al., 2020*; *Wu et al., 2020*; *Xiao et al., 2018*; *Xu et al., 2022*; *Yu et al., 2021*; *Zhang et al., 2019*). One study was from Japan (*Nagashio et al., 2010*) and the rest were from China (*Cao et al., 2010*; *Chen et al., 2015*; *Han et al., 2018*; *Jin et al., 2014*; *Li et al., 2016*; *Li et al., 2019*; *Lin et al., 2014*; *Liu et al., 2017*; *Liu et al., 2018*; *Lv et al., 2016*; *Peng et al., 2016*; *Qu et al., 2017*; *Shen et al., 2020*; *Shi et al., 2012*; *Wang et al., 2015*; *Wu et al., 2020*; *Xiao et al., 2018*; *Xu et al., 2022*; *Yu et al., 2021*; *Zhang et al., 2019*; *Zhong et al., 2013*). The thirteen studies used to assess the prognostic value included fifteen independent cohorts, with three studies for PC (*Han et al., 2018*; *Li et al., 2016*; *Zhang et al., 2019*), two studies each for GC (*Chen et al., 2015*; *Yu et al., 2021*) and lung cancer (LC) (*Qu et al.,*
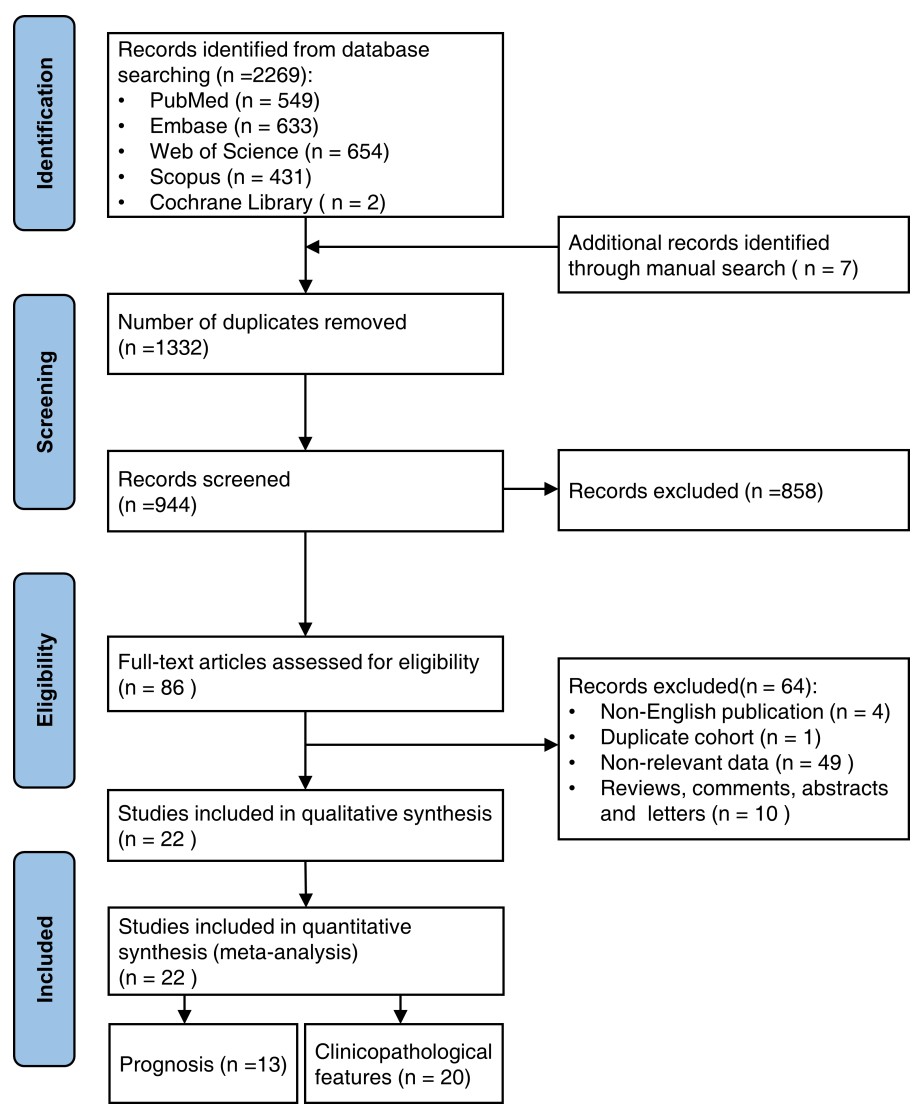

**Figure 1** **The flowchart showing the search and selection process of this study.**

*2017*; *Zhong et al., 2013*), one study with three independent cohorts for OSCC (*Liu et al., 2018*), and one study each for BC (*Cao et al., 2010*), glioma (*Lv et al., 2016*), ESCC (*Wang et al., 2015*), cervical cancer (CC) (*Wu et al., 2020*), and colorectal cancer (CRC) (*Xiao et al., 2018*), totaling 2,620 patients. The 20 studies used to evaluate the association between clinicopathological features and RACK1 expression included four studies for LC (*Nagashio et al., 2010*; *Qu et al., 2017*; *Shi et al., 2012*; *Zhong et al., 2013*), three studies each for GC (*Chen et al., 2015*; *Liu et al., 2017*; *Yu et al., 2021*), PC (*Han et al., 2018*; *Li et al., 2016*; *Zhang et al., 2019*), and CRC (*Jin et al., 2014*; *Li et al., 2019*; *Xiao et al., 2018*), two studies for CC (*Wu et al., 2020*; *Xu et al., 2022*), and one study each for BC (*Cao et al., 2010*), ovarian cancer (OC)(*Lin et al., 2014*), ESCC (*Wang et al., 2015*), melanoma (*Shen et al., 2020*), and nasopharyngeal carcinoma (NPC) (*Peng et al., 2016*), totaling 3,043 patients.

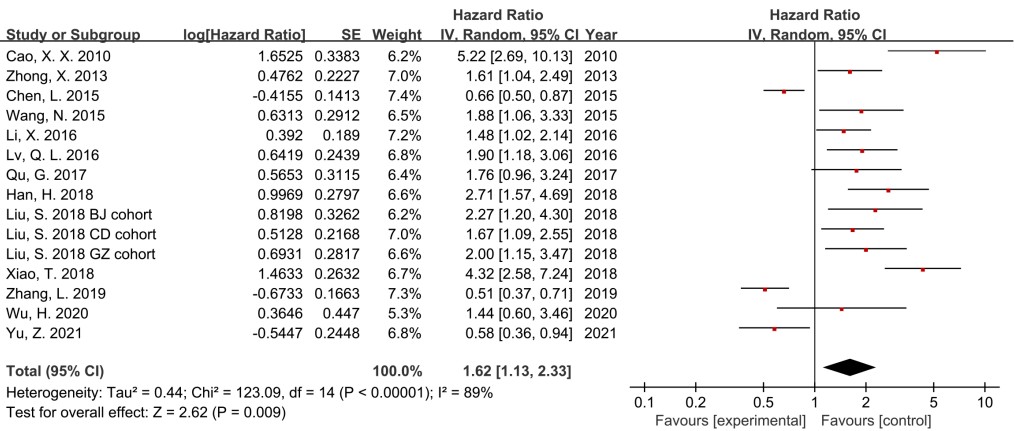

**Figure 2** Forest plot indicating the association between RACK1 expression and overall survival.

The sample sizes of the cohorts included in the meta-analysis ranged from 50 to 495 patients, with a mean of 147 patients. Two studies used reverse transcription-polymerase chain reaction (RT-PCR) to detect RACK1 expression, with immunohistochemistry (IHC) utilized in the remaining studies. The general characteristics of the included studies are shown in Table 1.

## Quality assessment

The overall quality of the included studies was good, with none of them scoring <6 (Table S3). Of the 22 studies evaluated, 11, 10, and 1 were individually scored 8, 7, and 6, respectively. The primary sources of bias include insufficient follow-up time, unclear cohort determination processes, and inconsistent interventions applied to the same cohort.

## Association between RACK1 expression and overall survival (OS)

Thirteen studies and 2,620 patients were included in the meta-analysis to assess the association between RACK1 expression and the prognosis of patients with cancer. The findings indicated that high RACK1 expression was significantly associated with poor OS despite the detection of significant heterogeneity (Fig. 2; HR = 1.62; 95% CI [1.13–2.33]; $P = 0.009$; $I^2 = 89\%$). Subgroup analysis, meta-regression, and sensitivity analysis were performed to address the observed heterogeneity.

## Association between RACK1 expression and disease-free survival/recurrence-free survival (DFS/RFS)

Two studies, including 192 patients, reported DFS or RFS data. Furthermore, the results demonstrated a significant correlation between high RACK1 expression and poor DFS/RFS, and no significant heterogeneity was indicated (Fig. 3; HR = 1.87; 95% CI [1.22–2.88]; $P = 0.004$; $I^2 = 0\%$).

Wang et al. (2023), *PeerJ*, DOI 10.7717/peerj.15873

**Table 1  General characteristics of each included literature in this systematic review and meta-analysis.**

| First author | Year | Country | Type of cancer | No. of patients (M/F) | Age | Detecting method | Treatment | Follow up (months) | Cut off | Survival information | TNM Stage | Source of HR and 95%CI | Clinicopathological characteristics |
|---|---|---|---|---|---|---|---|---|---|---|---|---|---|
| Cao, X. X. | 2010 | China | BC | 160 | 55.21(34–85) | IHC | Surgery | 72(1.5-108) | Scores >0 | OS | II-IV | R(M) | T, N, TNM, differentiation |
| Nagashio, R. | 2010 | Japan | LC | 123(68/55) | 65.5(41–85) | IHC | Surgery | NI | Scores ≥1 | NI | I-IV | NI | Gender, N, size |
| Shi, S. | 2012 | China | LC | 63(40/23) | NI | RT-PCR | Surgery | NI | NI | NI | I-IV | NI | Gender, age, N, TNM, differentiation, size |
| Zhong, X. | 2013 | China | LC | 180(123/57) | 60(37–75) | IHC | Surgery | 60 (3–96) | Scores ≥2 | OS | T1 | R(M) | Gender, age, N, differentiation |
| Jin, S. | 2014 | China | CRC | 157(76/81) | 58.8(30–85) | RT-PCR | Surgery | NI | Ratio of tumor/ pericarcinous tissu *e* > 1.15 | NI | I-IV | NI | Gender, N, differentiation |
| Lin, Y. | 2014 | China | OC | 50 | NI | IHC | NI | NI | Scores=2 or 3 | NI | I-IV | NI | M, TNM, differentiation |
| Chen, L. | 2015 | China | GC | 495(347/148) | 61.40(30–87) | IHC | Surgery | NI | NI | OS | I-IV | E | Gender, differentiation |
| Wang, N. | 2015 | China | ESCC | 100(79/21) | 60(42–78) | IHC | Surgery; surgery plus postoperative radiotherapy / chemotherapy / chemoradiotherapy | 49.5(3.0–71.0) | Scores >4 | OS, DFS | I-IV | R(M) | Gender, T, N, TNM, differentiation, size |
| Li, X. | 2016 | China | PC | 179(119/60) | 33-85 | IHC | Surgery | NI | Scores >4 | OS | I-IV | R(M) | Gender, age, N, TNM, differentiation, nerve invasion |
| Lv, Q. L. | 2016 | China | glioma | 92 | NI | IHC | Surgery | 48 | NI | OS | NI | R | NI |
| Peng, H. | 2016 | China | NPC | 58(41/17) | NI | IHC | NI | NI | NI | NI | I-IV | NI | Gender, T, M, TNM |
| Liu, C. | 2017 | China | GC | 70(43/27) | NI | IHC | Surgery | NI | Scores=12 | NI | I-IV | NI | Gender, age, TNM, differentiation |
| Qu, G. | 2017 | China | LC | 92(52/40) | 57.3 | IHC | Surgery | NI | Scores 2-3 | OS, RFS | I-IV | R(M) | Gender, N, TNM |
| Han, H. | 2018 | China | PC | 157(76/81) | 56(29–81) | IHC | Surgery | NI | Scores >0 | OS | I-IV | R(M) | Gender, age, N, TNM |
| Liu, S. | 2018 BJ cohort | China | OSCC | 83(65/18) | 60.89 ± 12.73 | IHC | Surgery; surgery plus radiotherapy and/or chemotherapy | 52 | Scores >6 | OS | I-IV | R(U) | NI |
| Liu, S. | 2018 CD cohort | China | OSCC | 151(107/44) | 61.07 ± 12.58 | IHC | Surgery; surgery plus radiotherapy and/or chemotherapy | 74 | Scores >6 | OS | I-IV | R(U) | NI |
| Liu, S. | 2018 GZ cohort | China | OSCC | 108(41/67) | 61.46 ± 12.45 | IHC | Surgery; surgery plus radiotherapy and/or chemotherapy | 78 | Scores >6 | OS | I-IV | R(U) | NI |
| Xiao, T. | 2018 | China | CRC | 180(100/80) | NI | IHC | Surgery | ≥70 | Scores 4-6 | OS | I-IV | E | Gender, N, TNM, differentiation |
| Li, X. Y. | 2019 | China | CRC | 205(120/85) | NI | IHC | NI | NI | Scores ≥3 | NI | I-IV | NI | Gender, N, M, TNM |
| Zhang, L. | 2019 | China | PC | 182(95/87) | NI | IHC | Surgery | NI | Scores ≥4 | OS | IA,IB,IIA,IIB | E | Gender, age, T, N, differentiation, size, nerve invasion |
| Shen, C. | 2020 | China | Melanoma | 67(40/27) | NI | IHC | NI | NI | Score 1-3 | NI | I-IV | NI | Gender, N, TNM |

Wang et al. (2023), *PeerJ*, DOI 10.7717/peerj.15873

**Table 1** (*continued*)

| First author | Year | Country | Type of cancer | No. of patients (M/F) | Age | Detecting method | Treatment | Follow up (months) | Cut off | Survival information | TNM Stage | Source of HR and 95%CI | Clinicopathological characteristics |
|---|---|---|---|---|---|---|---|---|---|---|---|---|---|
| Wu, H. | 2020 | China | CC | 306 | NI | IHC | Surgery | NI | Scores ≥7 | OS | FIGO I,II | E | Differentiation |
| Yu, Z. | 2021 | China | GC | 155(108/47) | NI | IHC | Surgery | NI | Scores 6-12 | OS | I-IV | E | Gender, T, N, TNM |
| Xu, L. | 2022 | China | CC | 104 | 49.52(28–68) | IHC | Surgery | NI | NI | NI | FIGO ≤IIB, >IIB | NI | N |

**Notes.**

BC, breast cancer; GC, gastric cancer; PC, pancreatic cancer; CC, cervical cancer; LC, lung cancer; CRC, colorectal carcinoma; OC, ovarian cancer; OSCC, oral squamous cell carcinoma; NPC, nasopharyngeal carcinoma; ESCC, esophageal squamous cell carcinoma; OS, overall survival; RFS, recurrence-free survival; DFS, disease-free survival; FIGO, The International Federation of Gynecology and Obstetrics; R(M), Data analyzed with multivariate Cox regression analysis were reported in study; R(U), Data analyzed with univariate Cox regression analysis were reported in study; E, HRs and 95%CI were estimated from Kaplan–Meier curves according to the method described by Tierney et al.; age, age ≥60 years old or age <60 years old; T, T stage; N, lymphatic metastasis; M, distant metastasis; TNM, TNM stage; size, tumor size (size ≥ 3 cm or size <3 cm); NI, not informed.

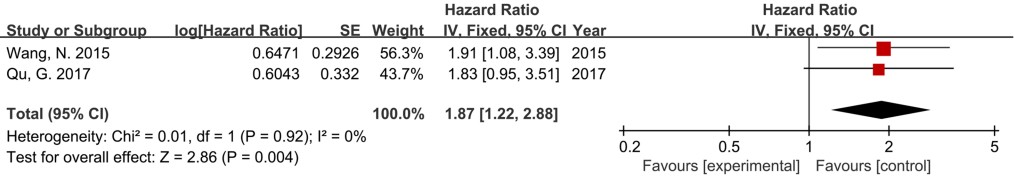

**Figure 3** Forest plot displaying the association between RACK1 expression and disease-free survival/recurrence-free survival.

**Table 2  The results of the association between RACK1 expression and clinicopathological characteristics.**

| Clinicopathological characteristics | Included studies | Included patients | Pooled OR (95% CI) | P | Heterogeneity | | Effect model |
|---|---|---|---|---|---|---|---|
| | | | | | $I^2$ (%) | P | |
| Age (≥60 vs. <60) | 6 | 831 | 1.74(0.88–3.42) | 0.11 | 73 | 0.002 | Random |
| Gender (male vs. female) | 16 | 2463 | 1.09(0.91–1.31) | 0.32 | 0 | 0.59 | Fixed |
| T stage (T3/T4 vs. T1/T2) | 5 | 655 | 0.98(0.35–2.79) | 0.97 | 84 | <0.001 | Random |
| Lymphatic invasion (N+ vs. N0)[a] | 15 | 2013 | 1.74(1.04–2.90) | 0.04 | 79 | <0.001 | Random |
| M stage (M+ vs. M0) | 3 | 260 | 1.27(0.36–4.47) | 0.71 | 66 | 0.05 | Random |
| TNM stage (IV/III vs. I/II) | 13 | 1483 | 1.58(0.85–2.94) | 0.15 | 80 | <0.001 | Random |
| Differentiation (poor vs. median/high) | 12 | 2099 | 1.56(0.84–2.86) | 0.16 | 85 | <0.001 | Random |
| Tumor size (≥3 cm vs. <3 cm) | 4 | 468 | 1.29(0.66–2.52) | 0.46 | 63 | 0.04 | Random |
| Nerve invasion | 2 | 361 | 0.74(0.18–3.07) | 0.67 | 89 | 0.003 | Random |

**Notes.**
[a]Data on lymphatic invasion was extracted from the N stage of TNM stages in the primary literatures.

## Association between RACK1 expression and clinicopathological characteristics

To evaluate the association between RACK1 expression and clinicopathological features, 20 cohorts containing 3,043 patients were included in this meta-analysis. The pooled results indicated that high RACK1 expression in patients with cancer was associated with lymphatic invasion/N+ stage (OR = 1.74; 95% CI [1.04–2.90]; $P = 0.04$; $I^2 = 79\%$) of tumors. No relationship was observed between RACK1 expression and age, gender, T stage, M stage, TNM stage, tumor differentiation, tumor size, or nerve invasion, as presented in Table 2 and Fig. S1. Significant heterogeneity was observed in each group when the effect sizes were combined, except when the relationship between gender and RACK1 expression was explored.

## Subgroup analysis and meta-regression

Subgroup analysis for RACK1 expression and OS was based on cancer type (digestive or non-digestive system cancers) and sample size (sample size ≥ 100 or <100), as well as the source of HRs(reported and estimated) and the Cox analysis method (multivariate and univariate). Except for the subgroups regarding the digestive system, estimated HR, and univariate Cox analysis, all other results of the subgroup analysis suggested that high RACK1 expression could predict poor OS (Table 3 and Fig. S2).

**Table 3  Subgroup analysis regarding association of RACK1 with overall survival and the results of meta-regression.**

| Subgroups | Included studies[a] | Included patients | Pooled HR (95% CI) | P | Heterogeneity | | Effect model | Meta-regression (P) |
|---|---|---|---|---|---|---|---|---|
| | | | | | I² (%) | P | | |
| **Cancer type** | | | | | | | | 0.327 |
| Digestive system | 10 | 1,790 | 1.44 (0.91–2.27) | 0.12 | 91 | <0.001 | Random | |
| Non-digestive system | 5 | 830 | 2.08 (1.38–3.13) | <0.001 | 59 | 0.04 | Random | |
| **Sample size** | | | | | | | | 0.604 |
| ≥100 | 12 | 2,353 | 1.55 (1.01–2.37) | 0.04 | 90 | <0.001 | Random | |
| <100 | 3 | 267 | 1.95 (1.41–2.69) | <0.001 | 89 | <0.001 | Random | |
| **Source of HR** | | | | | | | | 0.037 |
| Reported | 10 | 1,302 | 1.92 (1.64–2.25) | <0.001 | 34 | 0.14 | Fixed | |
| Estimated | 5 | 1,318 | 1.01 (0.50–2.03) | 0.98 | 93 | <0.001 | Random | |
| **Analyzing method** | | | | | | | | 0.189 |
| Multivariate | 6 | 868 | 2.09 (1.49–2.94) | <0.001 | 61 | 0.02 | Random | |
| Univariate | 9 | 1,752 | 1.35 (0.82–2.21) | 0.24 | 91 | <0.001 | Random | |

Notes.
[a] Study of Liu. S. et al. including 3 individual cohorts was considered as 3 separate studies for analysis.

**Table 4  The results of the association between RACK1 expression and lymphatic invasion/ N + stage regarding specific cancer types.**

| Cancer types | Included studies | Included patients | Pooled OR (95% CI) | P | Heterogeneity[a] | | Effect model[a] |
|---|---|---|---|---|---|---|---|
| | | | | | I² (%) | P | |
| Lung cancer | 4 | 420 | 2.36 (1.56–3.56) | <0.001 | 44 | 0.15 | Fixed |
| Colorectal carcinoma | 3 | 489 | 1.99 (0.66–6.02) | 0.22 | 83 | 0.003 | Random |
| Pancreatic cancer | 3 | 518 | 1.08 (0.40–2.92) | 0.88 | 85 | 0.002 | Random |
| Breast cancer | 1 | 160 | 10.42 (0.53–205.14) | 0.12 | N/A | N/A | N/A |
| Cervical cancer | 1 | 104 | 4.57 (1.42–14.74) | 0.01 | N/A | N/A | N/A |
| Esophageal squamous cell carcinoma | 1 | 100 | 2.46 (1.04–5.82) | 0.04 | N/A | N/A | N/A |
| Gastric cancer | 1 | 155 | 0.20 (0.08–0.48) | <0.001 | N/A | N/A | N/A |
| Melanoma | 1 | 67 | 1.52 (0.06–38.75) | 0.80 | N/A | N/A | N/A |

Notes.
[a] Due to the limited study on breast cancer, cervical cancer, esophageal squamous cell carcinoma, gastric cancer, and melanoma, the corresponding "heterogeneity" and "effect model" were not applicable (N/A).

Considering that different cancer types may have different propensities for lymphatic invasion, we analyzed the association between RACK1 expression and lymphatic invasion in specific cancer types. The findings indicated high RACK1 expression to be significantly associated with lymphatic invasion in LC (OR = 2.36; 95% CI [1.56–3.56]; $P < 0.001$; $I^2$ = 44%), but not statistically associated with PC and CRC. For other cancer types, definite conclusions could not be drawn because only one study for each could be included in the analysis (Table 4 and Fig. S3).

To further explain the heterogeneity among the studies focusing on RACK1 expression and OS, meta-regression was performed, and a P value <0.05 was utilized as the assessment standard for judging whether a certain factor serves as the source of heterogeneity. The findings implied that the source of HR ($P = 0.037$) might be a source of heterogeneity,

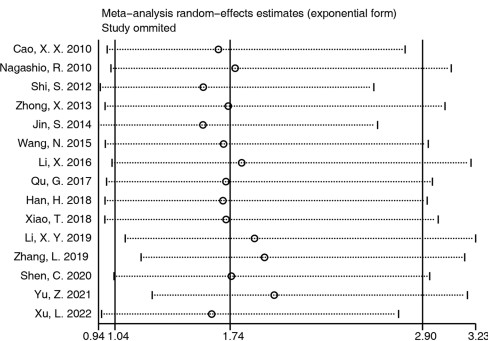

**Figure 4  Sensitivity analysis for meta-analysis of RACK1 expression and (A) overall survival and (B) lymphatic invasion/N+ stage.**

contributing 29.49% of the heterogeneity. The detailed results of the subgroup analysis and meta-regression are presented in Table 3.

## Sensitivity analysis

Sensitivity was evaluated to verify the stability of our conclusions when significant heterogeneity occurred in assessing the correlation of RACK1 expression with OS and lymphatic invasion. The sequential exclusion of each study did not affect the conclusion regarding the OS, indicating that the results were relatively stable and reliable (Fig. 4A). However, inconsistent combined results were observed for lymphatic invasion after removing specific literature; some pooled ORs indicated that RACK1 expression was not associated with lymphatic invasion (Fig. 4B). As a result, the conclusions regarding lymphatic invasion might be unstable.

## Publication bias

Visually inspecting the funnel plot, we found asymmetry in the OS analysis, and no asymmetry was found for DFS/RFS or lymphatic invasion (Fig. 5). Begg's and Egger's tests were performed to assess the funnel plot asymmetry. No publication bias was observed in the association between RACK1 expression and lymphatic invasion (Begg's test: $P = 0.692$; Egger's test: $P = 0.258$) of cancer. However, publication bias was found in the studies on the correlation between RACK1 expression and OS using Egger's test (Begg's test: $P = 0.092$; Egger's test: $P = 0.004$). As a result, caution was required when interpreting OS outcomes. No quantitative assessment of publication bias for DFS/RFS was conducted owing to insufficient literature.

## DISCUSSION

Numerous studies have investigated the relationship between RACK1 expression and the prognosis and clinicopathological characteristics of patients with cancer; however, conflicting conclusions have been drawn. To the best of our knowledge, this is the first systematic review and meta-analysis serving as a global proof that high RACK1 expression

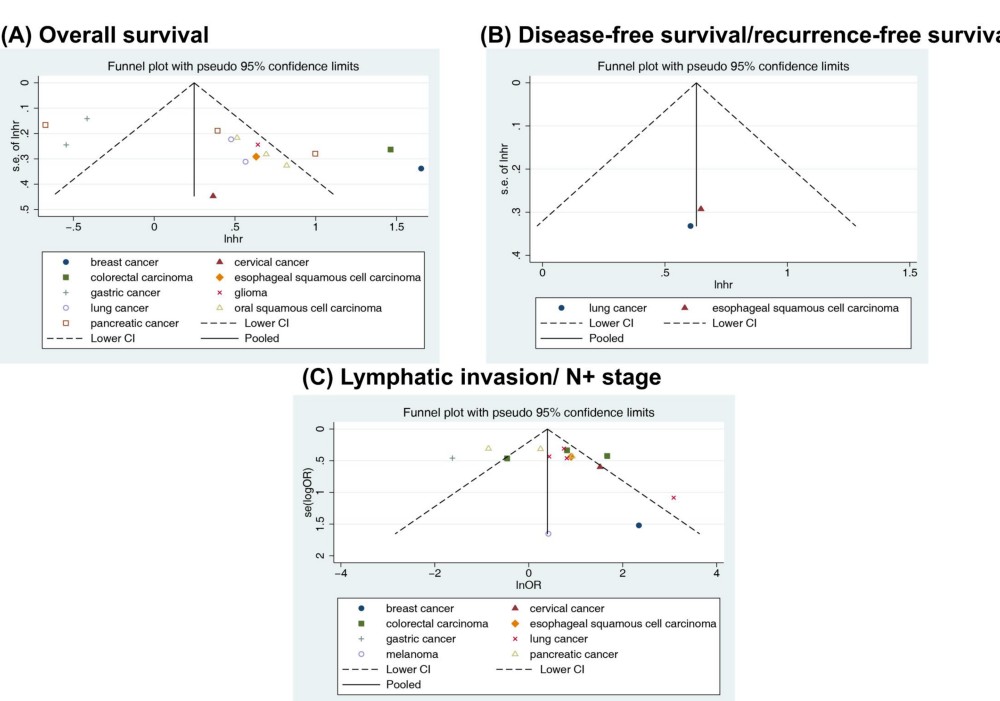

**Figure 5** Funnel plots with pseudo 95% confidence limits for the included studies reporting (A) overall survival, (B) disease-free survival/recurrence-free survival and (C) lymphatic invasion/N+ stage.

is a predictive marker of poor prognosis in various cancers, consistent with the findings of the majority of the studies. Increased RACK1 expression was significantly associated with lymphatic invasion, which might account for the poorer survival rate in RACK1-high patients with cancer.

In this study, thirteen studies investigating 2,620 patients with cancer were included to assess the relationship between RACK1 expression and OS, and the pooled HR indicated that higher RACK1 expression might predict poorer OS. The poor prognosis induced by RACK1 expression may be explained by the following mechanisms: first, in non-small cell lung cancer, RACK1 plays a key role in carcinogenesis by activating the AKT and FAK pathways (*Wu et al., 2021*); second, RACK1 might promote the development of esophageal cancer by activating the RAS/MEK/ERK pathway (*Li et al., 2022*); and third, RACK1 could accelerate the progression of OSCC *via* the AKT/mTOR pathway (*Zhang et al., 2016*). In contrast, *Chen et al. (2015)* reported that low expression of RACK1 may enhance the autocrine IL8 through miRNA-302c and contribute to an invasive or metastatic phenotype of gastric cancer, indicating the relationship between decreased RACK1 and poor prognosis in patients with GC; and *Zhang et al. (2019)* found that RACK1 down-regulation might be responsible for the occurrence of PC in its early stage by activating the NF-κB pathway (*Hu et al., 2019*). Therefore, organ specificity may contribute to the versatile roles of RACK1 in cancer, accounting for the heterogeneity of the pooled results from various cancers. In the subgroup analysis, except for the digestive system subgroups, estimated HR, and univariate Cox analysis method, all other subgroups confirmed that high RACK1 expression
was associated with a poorer prognosis. Many factors may account for the inconsistent results across studies, including, heterogeneity due to differences in cancers originating from different anatomic sites (as previously stated), distinct IHC results obtained under different conditions (such as staining on fresh or long-term stored sections or antibodies of varying natures), different percentages of tumor cells on various slides as RACK1 may also be expressed in non-tumor cells, affecting the scoring, and inclusion of a small sample size in the subgroup.

Recurrence is frequently a cause of disease deterioration and contributes substantially to cancer-related deaths in patients (*Fox et al., 2020*). Two studies and 192 patients were pooled to examine the relationship between RACK1 expression and DFS/RFS. The combined results demonstrated that high RACK1 expression was significantly and negatively associated with DFS/RFS, further strengthening the previous finding that RACK1 predicts a poor prognosis. Similarly, Wang et al. reported that high RACK1 expression was significantly associated with the recurrence of OSCC and predicted poor clinical outcomes (*Wang et al., 2009*), implying that clinicians should closely monitor the disease progression of patients with higher RACK1 expression and increase the frequency of follow-up visits if needed.

To evaluate the correlation between RACK1 expression and the clinicopathological characteristics of patients with cancer, 20 studies with 3043 patients were included. We found that cancer patients with high RACK1 expression exhibited a higher tendency for lymphatic invasion/N+ stage. These clinicopathological features may significantly contribute to poor survival outcomes in patients with cancer, consistent with the above finding that high RACK1 expression is associated with a poor prognosis (*Zhang et al., 2021*). Additionally, RACK1 expression was observed to be associated with the lymphatic invasion/N+ stage in LC. Even early-stage LC tends to invade the lymphatic system, contributing to its poor prognosis (*Sato et al., 2021*). During PC progression, lymphatic invasion could be observed early, despite the presence of only a few lymphatic vessels (*Fink, Steele & Hollingsworth, 2016*). CRC most commonly metastasizes to the liver *via* the bloodstream and to the lungs *via* the lymphatic system (*Naxerova et al., 2017*). Notably, RACK1 may promote the migration and invasion of malignant tumors by mediating the epithelial-mesenchymal transition in LC, the PI3K/Akt pathway in PC, and the AMPK/YAP pathway in CRC (*Kong et al., 2020*; *Qu et al., 2017*; *Sun et al., 2021*). However, its expression did not correlate with lymphatic invasion in CRC and PC in our study; therefore, more well-designed studies are required to ascertain the role of RACK1 in various cancers. Apart from the lymphatic invasion/N+ stage, no correlation between RACK1 and the remaining clinicopathological features (age, gender, T stage, M stage, TNM stage, differentiation, tumor size, and nerve invasion) was observed, implying that RACK1 might contribute to the poor prognosis of patients with cancer primarily by promoting lymphatic invasion.

This study had some limitations. First, the patients with cancer included in the literature were all from East Asia; this may result in bias considering the homogeneity of the region and ethnicity. Second, no uniformity in determining the cut-off value of high/low RACK1 expression was observed; additionally, subjectivity in scoring IHC findings could not be ignored, and inconsistency regarding the selection of antibodies for IHC or primers for

RT-PCR among the included original studies might compromise the validity of the pooled effect sizes. Third, only a limited number of studies (one each for LC and ESCC) were considered for DFS/RFS, which might have resulted in non-convincing results. Fourth, the conclusion regarding lymphatic invasion was found to be unstable by sensitivity analysis, and potential publication bias was observed when exploring the relationship between RACK1 and OS, potentially diminishing the benefit of this meta-analysis. In the future, more well-designed studies are warranted to validate the value of RACK1 as a prognostic factor for cancer compared to other recognized typical markers.

## CONCLUSION

Our systematic review and meta-analysis demonstrated from an evidence-based medical perspective that RACK1 is a relatively global marker of poor prognosis in patients with cancer; particularly, high RACK1 expression was indicated to be significantly associated with poorer survival (OS and DFS/RFS) and worse clinicopathological characteristics (lymphatic invasion/N+ stage). As a result, patients with high RACK1 levels should have more frequent follow-ups to avoid unfavorable clinical outcomes. Owing to the limited number of studies and types of cancers included in this review, more high-quality studies are needed to validate the findings of our study.

### Funding

This work is supported by the National Natural Science Foundation of China (82272899, 81902782, 82203180), the Research Funding from West China School/Hospital of Stomatology Sichuan University (No. RCDWJS2022-16), the Innovation and Entrepreneurship Training Scheme for University Students Program of Sichuan University (20231525L), the 14th special grant from China Postdoctoral Science Foundation (2021T140484), the Postdoctoral Research Funding of Sichuan University (2022SCU12132), the Research and Exploration Program of West China Hospital of Stomatology of Sichuan University (No. RD-02-202204), and the Key Research Program of Sichuan Provincial Science and Technology Agency (2023YFS0127). The funders had no role in study design, data collection and analysis, decision to publish, or preparation of the manuscript.

### Grant Disclosures

The following grant information was disclosed by the authors:
National Natural Science Foundation of China: 82272899, 81902782, 82203180.
Research Funding from West China School/Hospital of Stomatology Sichuan University: RCDWJS2022-16.
Innovation and Entrepreneurship Training Scheme for University Students Program of Sichuan University: 20231525L.
14th special grant from China Postdoctoral Science Foundation: 2021T140484.

Postdoctoral Research Funding of Sichuan University: 2022SCU12132.
Research and Exploration Program of West China Hospital of Stomatology of Sichuan University: RD-02-202204.
Key Research Program of Sichuan Provincial Science and Technology Agency: 2023YFS0127.

## Competing Interests

The authors declare there are no competing interests.

## Author Contributions

- Qiuhao Wang performed the experiments, analyzed the data, prepared figures and/or tables, authored or reviewed drafts of the article, and approved the final draft.
- Sixin Jiang performed the experiments, analyzed the data, prepared figures and/or tables, authored or reviewed drafts of the article, and approved the final draft.
- Yuqi Wu performed the experiments, prepared figures and/or tables, authored or reviewed drafts of the article, and approved the final draft.
- You Zhang performed the experiments, analyzed the data, prepared figures and/or tables, authored or reviewed drafts of the article, and approved the final draft.
- Mei Huang analyzed the data, prepared figures and/or tables, authored or reviewed drafts of the article, and approved the final draft.
- Yan Qiu conceived and designed the experiments, authored or reviewed drafts of the article, and approved the final draft.
- Xiaobo Luo conceived and designed the experiments, authored or reviewed drafts of the article, and approved the final draft.

## Data Availability

   Raw data are available in the Supplemental Files.

## Supplemental Information

Supplemental information for this article can be found online at http://dx.doi.org/10.7717/peerj.15873#supplemental-information.

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
