# Peer review of "Prognostic and clinicopathological role of RACK1 for cancer patients: a systematic review and meta-analysis"

_PeerJ, doi:10.7717/peerj.15873_

## Round 0.1 · original submission · Minor Revisions

Please address the issues pointed out by the reviewer and amend the manuscript accordingly.

Reviewer 1 ·

Basic reporting

line 103, typo "the opposite finding is revealed in PC"

Experimental design

about the part "Association between RACK1 expression and clinicopathological characteristics". The patient cohort used here includes pan-cancer, but not all cancer types tend to metastate through lymphatics. I wonder whether the authors take this factor into consideration. I think a subgroup analysis could be done to make the conclusion "high
213 RACK1 expression in cancer patients were associated with lymphatic invasion" more precise and cancer-type specific.

In addition, the author didn't clarify how lymphatic invasion is primarily defined in the studied literature. If the information comes from TMN stages, then the N stage should be also shown to correlate with RACK1 expression.

Validity of the findings

No

Additional comments

This systematic review and meta-analysis provided an overview of studies about the association between RACK1 expression and clinical features and outcomes. It implies that RACK1 expression negatively correlates with cancer patient survival and seems to associate with lymphatic metastasis.

The article is overall well-written and could be published after addressing the comments above.

Reviewer 2 ·

Basic reporting

Authors Wang et. al. have performed an extensive analysis of RACK1 literature available in several scientific databases to perform meta-analysis and systemic review. The manuscript describes the correlation of high expression of RACK1 with overall survival and disease-free survival/recurrence free survival. The analysis presented that the high expression of the RACK1 is associated with low overall survival rates in cancer patients regardless of their age, T stage, M stage, TNM stage, gender, differentiation, tumor size and nerve invasion.
This review also highlights the limitations of the meta-analysis suggesting more robust, well-designed, and newer studies are required to further advance the meta-analysis to cover wide range of parameters such as race of the patients, standard cut-off for RACK1 expression level, wide range of cancers etc.
Overall, the manuscript is well written and is accurate grammatically. The analysis of literature by using inclusion and exclusion criteria is accomplished in clear manner. The authors have provided reasonable data to support the hypothesis. The figures and tables are included where necessary and are in proper format.

Experimental design

The manuscript demonstrates several statistical methods to support the hypothesis. The methods used are Newcastle-Ottawa scale and Kaplan-Meier curves to statistically analyze the data obtained from the literature. Proper explanation and references are provided to reproduce the data.

Validity of the findings

The data obtained through statistical analysis is validated using values such as P value and confidence intervals (CI%). Hazard ratios (HR) and odd ratios (OR) are used along with CIs to evaluate the correlation of RACK1 expression. The bias is eliminated or reported where necessary. The selection of data is done using defined inclusion and exclusion criteria. The discussion and conclusion section describes the findings in a clear and concise manner. The advantages and limitations of the meta-analysis are elaborated and suggestions for future improvements are also provided.

Additional comments

There is a small typing error in line 103 where “he” is written instead of “the”?
Table 1 is hard to read and would be nice if it is in horizontal orientation which will make it easier to read.

Reviewer 3 ·

Basic reporting

The authors performed a systematic meta-analysis on the prognostic value of RACK1 expression in cancers and its correlation with patients clinicopathological features. Indeed, the protein RACK1 has been shown to be associated to cancer outcome, although with conflicting reports. The protocol is well described and was registered.
The article is written in the perspective of a clinician, far from the scientist perspective. The introduction does not consider the extensive although imperfect knowledge in the RACK1 field considering for example the ubiquitous expression of RACK1 and its high expression in normal epithelia. This may produce a bias in the interpretation of results.
Reference 8 appears as 2015 but Pubmed cites it as 2016 (PeerJ. 2016 Feb 16;3:e1499. doi: 10.7717/peerj.1499. eCollection 2015)

Figure 5 could include the labels of A, B and C panels on each panel and “Funnel plot with pseudo 95% confidence limits” could stay in the legend. A color code on dots to identify cancer types in the Funnel plots could be interesting.

The manuscript is clear and well written.

Experimental design

The classic experimental design comes up against the comparison of studies using detection methods as immunohistochemistry or RT-PCR. In both cases, an important bias on the choice of the antibodies and the primers may occur and comparing results on IHC without in vitro diagnostic standardised antibodies might not be relevant. Unfortunately, in most of the articles the reference of the antibody is not provided.
It would be interesting to know how RACK1 prediction works compared to other more typical markers in these cancers in these same cohorts. But this might be another paper.

Validity of the findings

The conclusions seem to disregard the limitation the authors have well identified. They should replace “convincing proof that high RACK1 expression” (lane 253) by for instance “global proof” to show that it does not universally apply. Same in lane 306 “RACK1 is a potent marker of poor prognosis”.
They raise the small sample size of the studies, but multiple reasons could explain the partial inconsistency of the findings, as for the heterogeneity of the cancer lesions types, the IHC technique which might work differently on fresh or long-term stored sections, the nature of the antibodies, or the percent of tumor cells in the selected lesions as the microenvironment may also express high levels of RACK1.

---

## Round 0.2 · accepted · Accept

All issues pointed by the reviewers were adequately addressed and amended manuscript is acceptable now.